# Automatic Weight Redistribution Ensemble Model Based on Transfer Learning to Use in Leak Detection for the Power Industry

**DOI:** 10.3390/s24154999

**Published:** 2024-08-02

**Authors:** Sungsoo Kwon, Seoyoung Jeon, Tae-Jin Park, Ji-Hoon Bae

**Affiliations:** 1Department of AI and Big Data Engineering, Daegu Catholic University, 13-13, Hayang-ro, Hayang-eup, Gyeongsan-si 38430, Republic of Korea; sungsoo@cu.ac.kr (S.K.); jeon1008sy@cu.ac.kr (S.J.); 2Nuclear System Integrity Sensing & Diagnosis Division, Korea Atomic Energy Research Institute (KAERI), 62, Gwahak-ro, Yuseong-gu, Daejeon 34142, Republic of Korea; etjpark@kaeri.re.kr

**Keywords:** leak detection, artificial intelligence, transfer learning, ensemble model, ensemble weight automatic redistribution

## Abstract

Creating an effective deep learning technique for accurately diagnosing leak signals across diverse environments is crucial for integrating artificial intelligence (AI) into the power plant industry. We propose an automatic weight redistribution ensemble model based on transfer learning (TL) for detecting leaks in diverse power plant environments, overcoming the challenges of site-specific AI methods. This innovative model processes time series acoustic data collected from multiple homogeneous sensors located at different positions into three-dimensional root-mean-square (RMS) and frequency volume features, enabling accurate leak detection. Utilizing a TL-driven, two-stage learning process, we first train residual-network-based models for each domain using these preprocessed features. Subsequently, these models are retrained in an ensemble for comprehensive leak detection across domains, with control weight ratios finely adjusted through a softmax score-based approach. The experiment results demonstrate that the proposed method effectively distinguishes low-level leaks and noise compared to existing techniques, even when the data available for model training are very limited.

## 1. Introduction

Pipelines installed during the initial construction phase of domestic and international power plants are now facing increasingly serious leak issues caused by aging. Such leaks threaten the safety and efficiency of normal power plant operations, leading to potential and continuous operational disruptions. Economic losses and environmental pollution caused by leaks are serious social issues that also give rise to various side effects in power plant operations. Therefore, the early detection of leaks in aging pipelines and the appropriate responses are essential for the continuous operation of power plants. To this end, there is an emphasized need to introduce advanced technologies and establish a regular inspection system. Traditionally, acoustic, vibration, and ultrasound sensors have been primarily used to detect pipeline leaks [1,2,3,4,5]. Lee et al. [6] developed an ultrasonic leak detector for remotely detecting pipeline leaks using noncontact and nondestructive methods. Meanwhile, Li et al. [7] analyzed the location and error of gas pipe leaks using acoustic emission (AE) sensors. However, these methods have limitations in distinguishing between leak signals and noise in situations that include various mechanical noises. To address these limitations, research has been conducted to differentiate between leak and noise signals by finding and compressing distinguishable features from acoustic signals collected by remote microphone sensor nodes, or by sparsifying signals and removing noise to differentiate leak signals using wavelet transform techniques applied to acoustic leak detection signals [8,9,10].

Recently, with the advent of the big data era and the rapid development of deep learning and machine learning technologies, research on solving field problems using industrial data is actively progressing. Studies utilizing artificial intelligence have offered new solutions for various situations and have significantly contributed to improving industrial productivity and efficiency. Among these, the use of artificial intelligence in detecting microleaks in aging pipelines is receiving considerable attention. As an example in this field, Salam et al. [11] conducted research using a single-mode optical-fiber (SMF-28)-based acoustic vibration sensor and a support vector machine (SVM) to predict the location and size of leaks. In addition, various studies using machine learning techniques like SVM for leak detection are ongoing [12,13,14,15].

The advancements in deep learning methodologies also suggest their potential applicability in leak detection research [16,17,18,19,20,21]. Wang et al. [22] applied leak acoustic signals transformed into the frequency domain via fast Fourier transform (FFT) to artificial neural networks for leak detection, achieving a recognition accuracy of 96.87%. Then, C. Spandonidis, C. et al. [23] applied spectrograms extracted from signals obtained from accelerometers to two-dimensional (2D) convolutional neural network (CNN) and long short-term memory autoencoder (LSTM-AE) models, achieving 92% accuracy in leak detection. Zhou et al. [24] proposed a one-dimensional (1D) CNN ensemble model that applies particle swarm optimization to ensemble weight optimization for leak detection, showing a precision of 90.5%. However, such studies have shown limitations in accurately detecting low-level leaks. In addition, there is an emphasis on the need for research that demands high performance even with limited datasets because of the small amount of data available.

This paper proposes a transfer-learning (TL)-based deep learning model incorporating automatic weight redistribution for feature fusion from multiple sensors to detect low-level leaks in pipelines. Initially, independent features are extracted from the time and frequency domains for leak and noise signals measured by multiple homogeneous sensors positioned at various locations. These features are then combined within each domain to create volume features. First, 2D RMS pattern features are extracted from the time series signals of each sensor in the time domain, and 2D frequency pattern features are extracted using the same time series signals in the frequency domain. Second, the 2D pattern features extracted from multiple sensors in each domain are combined in the depth direction of the 3D axis to create RMS volume features and frequency volume features independently. These RMS and frequency volume features are ultimately used to train the TL-based ensemble deep learning model proposed in this research.

Next, the approach to applying TL in training the leak detection ensemble model consists of the following two stages. In Stage 1, the training of single models based on the residual network (ResNet) [25] in each domain is designed and completed, after which the models are saved. These saved models are a preparatory step for applying TL in Stage 2. In Stage 2, the pretrained models from each domain in Stage 1 are transferred to perform model ensembling.

At this point, a method that automatically redistributes ensemble weights for each domain in the ensemble model was proposed, thereby further enhancing the accuracy of leak detection based on TL. Here, the ensemble method can achieve better performance by combining several models compared to a single model alone. It also reduces overfitting and improves generalization performance, significantly increasing the model’s prediction accuracy and reliability [26,27]. TL is beneficial when there is a lack of sufficient training data for a new dataset, as it utilizes the weights and structure of previously trained models. Furthermore, it has the advantage of transferring knowledge acquired from large datasets to other related tasks, enabling the maintenance of high performance even with limited data.

The following are the main contributions of this study.

(1)Microleak detection is a significant problem; a phased and systematic approach is needed to address it effectively. In Stage 1, key features are extracted from time series signals in each independent domain. Pattern features from time and frequency domains are extracted from data collected by multiple sensors, and features from each domain are combined for each sensor, presenting a method for generating RMS and frequency volume features suitable for deep-learning-based ensemble learning.(2)In the ensemble learning process, this study introduced an innovative approach that automatically redistributes ensemble weights among models, in contrast to conventional ensemble techniques that rely on manual weight adjustment. This automatic redistribution mechanism allows each model within the ensemble to achieve optimal performance by dynamically adjusting weights based on data characteristics during the learning process. Consequently, this approach effectively improves the accuracy and efficiency of leak detection in an end-to-end manner, ensuring that the various models work synergistically to achieve superior overall performance.(3)An ensemble model integrated with TL was constructed to learn features extracted from two distinct domains effectively. The application of the TL technique ensures the stability and high performance of the leak detection ensemble model, even when working with a limited dataset.

The remainder of this paper is organized as follows. Section 2 explains the data preprocessing methods for feature extraction in each domain introduced above. Section 3 provides a detailed step-by-step description of the TL-based automatic weight redistribution leak detection ensemble model proposed in this paper. Section 4 presents the performance results and analysis of the proposed model. Section 5 concludes the paper and discusses future research directions.

## 2. Data Preprocessing for the Proposed Method

This section describes data preprocessing techniques for transforming 1D time series acoustic signals collected from multiple microphone sensors into a format like 2D images. This transformation was intended to facilitate training CNN-based deep learning models for distinguishing between leaks and normal conditions. For this, the method proposed in [28] was utilized to extract an RMS pattern feature representing magnitude information from the 1D time series signals and a frequency pattern feature from the frequency domain corresponding to the time domain, as shown in Figure 1.

The time series signal of x is fed into two separate signal processing paths, Path A and Path B, to create a 2D RMS pattern feature. In advance, this process performs input signal sampling at a $T_{s}$ sampling interval, such as $x_{n}=x(nT_{s})$. Afterward, in Path A, the RMS magnitude of the sampled input data is directly calculated as $y_{a}={\left[y_{n}\right]}_{n=1,2,\ldots ,T}$ using (1) without any digital filtering.
(1)$$y_{n}=\frac{1}{A}\sqrt{\sum_{i=1}^{W_{n}-1}x_{n+i}^{2},}\;n=1,2,\ldots ,T,$$
where A represents the normalization coefficient, and $W_{n}$ represents the window size for averaging the square root of the received data.

In Path B, the RMS magnitude is determined by filtering the desired signal region from the sampled input data. The band-pass filter (BPF) block is employed to achieve this, allowing only the specific band where the leak signal is distributed to pass while blocking other signals. This results in $y^{\prime }$, which has passed through the BPF. Subsequently, $y^{\prime \prime }$ is calculated, as described in (2), by emphasizing specific signals within the band-pass of the filter using the amplitude weighting block.
(2)$$y^{\prime \prime }=IFT\left[W(f)\cdot FT\left[y^{\prime }\right]\right],\;$$
where $IFT\left[\cdot \right]$ and FT[·] represent the 1D inverse Fourier transform and Fourier transform (FT), respectively. At this time, the window W(f) serves to amplify the signal with a concentrated spectrum to emphasize the leak signal and reduce unwanted external noise, as shown in (3).
(3)$$W\left(f\right)=\left\{\begin{array}{l} \alpha ,\;f_{L}\leq f\leq f_{H}\\ 0,\;otherwise, \end{array}\right.$$
where, $f_{L}$ and $f_{H}$ represent the start and stop frequencies of the window operation, respectively, and α denotes the signal amplification weight in the given frequency range. Subsequently, $y^{\prime \prime }$ calculated from (2) is inputted into the RMS block to generate $y_{b}$.

The two RMS-level feature vectors, $y_{a}$ and $y_{b}$, generated through Path A and Path B, respectively, are concatenated as in (4) to represent two inherent RMS information sets ($y_{rms}$).
(4)$$y_{rms}=\left[y_{a},y_{b}\right].$$

The transformed RMS feature vector from (4) is inputted into the quantization block of Figure 1, where it undergoes quantization into *N*-levels of magnitude. Subsequently, the 2D domain mapping block is divided into M time sample units, and the average of the quantized magnitude levels for each grid is calculated, resulting in the final M×N RMS pattern features as shown in Figure 2.

As shown in Figure 2a, the generated RMS pattern features are relative to the center of the 2D domain, with the left side representing the unfiltered portion of Path A and the right side representing the filtered portion of Path B. The leak signal after filtering shows a pattern rise in the right portion because of the applied weighting. In contrast, the noise signal, not being weighted, exhibits a pattern decline in the right portion.

The same time series signal used for RMS pattern feature extraction generates frequency pattern features, producing spectral magnitudes in the frequency domain. First, the time series acoustic data are passed through a BPF block. Second, the are transformed into frequency domain data using the 1D Fourier Transform (FT), as described in (5), and the spectral magnitude is calculated by taking the absolute value.
(5)$$\left|Y(f)\right|=\left|FT(y^{\prime })\right|.$$

The transformed frequency feature vector from (5) is entered into the quantization block, as depicted in Figure 1, similar to the 2D RMS pattern feature generation process, to perform *N*-level quantization. Subsequently, it passes through the 2D domain mapping block to generate the final M×N frequency pattern features, as illustrated in Figure 2b.

The generated 2D RMS pattern features and 2D frequency pattern features are combined with domain features extracted from multiple sensors for learning purposes. As shown in Figure 3, these pattern features are aligned in the depth direction using four sensors sequentially to form RMS and frequency volume features. Consequently, these volume features, which amalgamate the extracted features from several sensors in each area, are ultimately employed in the ensemble learning of the leak detection model proposed in Section 3 to represent the characteristics of leak signals effectively.

## 3. Proposed Method

This section describes the structure of the TL-based automatic weight redistribution ensemble model for fine leak detection, as shown in Figure 4. The proposed structure consists of two stages, and the application of TL is performed between Stages 1 and 2. Typically, TL [29,30] is employed to extract rich knowledge from a pretrained model in the source domain. Furthermore, TL aids in preventing overfitting, especially when the amount of training data for the target task is limited.

### 3.1. Stage 1 of the Proposed Method

Figure 5 illustrates the single model architecture of Stage 1, the initial phase of the TL-based automatic weight redistribution ensemble model. In Stage 1, a ResNet deep learning model with residual connections based on a 2D CNN [31] was employed. This model is specifically trained for classifying low-level leak signals and noise signals, utilizing RMS and frequency volume features obtained from each domain. The model is designed to extract high-dimensional features from the input data through multiple convolutional layers, and each layer is tailored to learn the spatial characteristics of the input data using 2D convolution operations.

ResNet is a model that enhances the training of deep neural networks by incorporating residual connections. It mitigates the problem of gradient vanishing, which can occur during the model training process, by introducing a shortcut connection [25] through identity mapping to the output of the function $F\left(x\right)$ for the input x, as shown in (6).
(6)$$y=F\left(x,\;\left\{W\right\}\right),$$
where x represents the feature map input to residual learning, while F(x) denotes a function comprising pairs of convolutional layers and batch normalization [32] layers that include weight parameters for training. The convolution operation in the 2D convolutional layer l, which makes up the function F(x) used for residual learning, is defined as in (7) based on the structure shown in Figure 6.
(7)$$x_{\left(i,j,n\right)}^{l}=h_{l}\left(\sum_{m\in R_{n}^{l}}\left[x_{\left(i,j,n\right)}^{\left(l-1\right)}*w_{\left(i,j,\left(m,n\right)\right)}^{l}\right]+b_{\left(i,j,n\right)}^{l}\right),\;\;n=1,2,\ldots ,N_{i}^{l}\;and\;l=1,2,\ldots ,L,$$
where $x_{\left(i,j,n\right)}^{l}$ represents the 2D feature map n in the current layer l, $x_{\left(i,j,m\right)}^{\left(l-1\right)}$ represents the 2D feature map m in the previous layer $\left(l-1\right)$, $w_{\left(i,j,\left(m,n\right)\right)}^{l}$ represents the kernel function of size $H_{c}^{l}\times W_{c}^{l}$ for the convolution operation from the 2D feature map m in layer (l−1) to the 2D feature map n in layer l, $b_{\left(i,j,n\right)}^{l}$ represents the bias term in the current layer l, and $h_{l}(\cdot )$ represents the activation function in layer l. At this time, i and j represent the spatial pixel indices of the feature map, $N^{l}$ represents the number of feature maps produced in layer l, and L represents the total number of layers where convolution operations are applied. In this context, padding techniques, which fill the edges of the input feature map with zeros to maintain the same spatial size of the feature map before and after the convolution operation, are applied. The widely used rectified linear unit (ReLU) [33] in deep learning training was employed for the activation function.

Therefore, configuring a single residual block structure, as shown in Figure 5, which comprises pairs of consecutively connected 2D convolutional layers and batch normalization layers, enables the automatic learning of meaningful high-dimensional features from the preprocessed volume feature data. In turn, this configuration facilitates the effective classification of leaks and noise.

For feature extraction of leak and noise signals, each domain-specific ResNet model in this study comprised two residual blocks, as illustrated in Figure 5. In the first residual block, convolutional layers were configured to output 16 feature maps using a 3 × 3 kernel filter. This kernel size was selected for the 40 × 40 × 4 input data due to its efficacy in capturing spatial features while maintaining computational efficiency. In addition, the 16 feature maps in the first residual block were deemed sufficient to extract essential features in the initial stage, mitigating risks of overfitting and excessive computational load.

The second residual block’s convolutional layers were configured to output 32 feature maps, enabling the capture of more complex patterns in deeper network layers. This progressive increase in feature map quantity allowed the model to learn more detailed and abstract representations, thereby enhancing its capacity to differentiate between leak and noise signals. While larger kernel sizes were considered, they were rejected due to their higher computational cost and potential for overfitting. Conversely, smaller kernels were found inadequate for capturing the requisite contextual information.

Within each residual block, batch normalization was applied post-convolution to address the internal covariate shift problem, with ReLU employed as the activation function. Global average pooling (GAP) was applied to the final residual block’s output feature map to compute the mean of all the feature values. This result was subsequently transformed into a 1D vector to extract key features. Finally, to classify leak and noise signals from the 1D feature vector, two fully connected layers with 32 and 2 neurons, respectively, were introduced, and a softmax function was applied to the output layer.

### 3.2. Stage 2 of the Proposed Method

In Stage 2, pretrained models from two domains independently trained and stored in Stage 1, are loaded. The classifier of each loaded model is removed and replaced with new linear classifiers. Subsequently, an ensemble retraining is conducted for both models, as illustrated in Figure 4. Thus, in Stage 2, the classifiers from the existing models in Stage 1 are excluded, and the pretrained knowledge derived from the base structure of CNNs, which facilitated extracting salient features for discriminating between leak and noise signals, is utilized.

Specifically, the residual block structure used in Stage 1 is retained in Stage 2. These residual blocks enable the deep neural network to learn effective high-dimensional representations, allowing the features learned in Stage 1 to be efficiently used in Stage 2. The high-dimensional features acquired through the residual blocks are crucial for distinguishing leak and noise signals. Furthermore, the GAP and fully connected layer structure applied to the output of the final residual block in Stage 1 is also retained in Stage 2. GAP computes the average of all the feature values and transforms them into a 1D vector, enabling the newly applied linear final classifier to classify the signals effectively.

Thus, through this process, the model pretrained in Stage 1 can achieve more sophisticated feature extraction and classification performance for new leak and noise signals in Stage 2 through TL. During this TL-based model ensemble training, the ensemble weights $\alpha_{t}$ and $\alpha_{f}$ for each model are not manually inserted. Instead, they are automatically distributed to each model during the learning process through the “Ensemble Weight Redistribution Block,” as presented in Figure 4. The detailed structure of this “Ensemble Weight Redistribution Block” is provided in Figure 7.

To extract ensemble weights, initially, the maximum value $s_{k}^{max}$ among class scores is extracted from the vector ${\bar{s}}_{k}=\left[s_{k}^{1},s_{k}^{2},\ldots ,s_{k}^{n}\right]$ produced from the output layer of each model classifier, as indicated in (8).
(8)$${s_{k}^{max}=\underset{n}{\max}\left(s_{k}^{1},s_{k}^{2},\ldots ,s_{k}^{n}\right)|}_{k=1,\ldots K,}$$
where k represents the index of the k-th domain among the total K domains, and $s_{k}^{n}$ denotes the score of the n-th class outputted from the output layer of the ResNet model for the k-th domain. Subsequently, before generating ensemble weights using $s_{k}^{max}$, as described in (9), $s_{k}^{max}$ is associated with a trainable weight parameter $w_{k}$ and bias parameter $b_{k}$, and batch normalization is applied. This formula creates reliable ensemble weights, allowing stable training and parameter adjustment in the k-th domain model.
(9)$$s_{k}^{BN}={BN\left(w_{k}\cdot s_{k}^{max}+b_{k}\right)|}_{k=1,\ldots K,}$$
where BN means batch normalization. In this way, the maximum scores $s_{k}^{BN}$ computed after normalization in each domain are sequentially connected using the $f^{cat}$ function in (10) to form a single 1D vector ${\bar{s}}^{cat}$.
(10)$${\bar{s}}^{cat}=f^{cat}\left(s_{1}^{BN},\;s_{2}^{BN},\;\ldots ,\;s_{K}^{BN}\right).\;$$

Next, a softmax function is applied to ${\bar{s}}^{cat}$ as described in (11), ensuring that the ensemble weights distributed to each domain model are calculated in proportion to the normalized maximum scores $s_{k}^{BN}$.
(11)$$\alpha_{k}={softmax\left({\bar{s}}^{cat}\right)|}_{k=1,\ldots K.}$$

Therefore, the ensemble weights $\alpha_{k}$ calculated in (11) are applied to the original output vector ${\bar{s}}_{k}$ of the k-th domain model, ensuring that the ensemble model training is ultimately performed feedforward.
(12)$$O_{p}=\sum_{k=1}^{K}\alpha_{k}\cdot {\bar{s}}_{k}$$

Ultimately, the ensembled $O_{p}$ in (12) is processed to distinguish between leak and noise by extracting the final class probability distribution through a softmax function, as depicted in Figure 7, thereby progressing the training of the ensemble model. In this study, considering two domains, K is set to 2 and, using (11), the ensemble weight applied to the time domain model is denoted as $\alpha_{t}$, and that applied to the frequency domain model as $\alpha_{f}$. The complete process and algorithm for training the automatic redistribution model of ensemble weights applied with TL in Stages 1 and 2 are shown in Figure 8.

## 4. Experimental Results and Analysis

### 4.1. Experimental Setup for Data Acquisition

In this research, for artificial-intelligence-based leak detection, a plant piping specimen, as depicted in Figure 9, and four commercial microphone acoustic sensors were installed to collect leak and noise signal data. The commercial microphone acoustic sensors used were the HT378A06 Integrated Circuit Piezoelectric (ICP) microphone from PCB Piezotronics [34]. To mitigate shadow areas and eliminate blind spots during leak detection, four commercial microphone acoustic sensors were strategically positioned at each corner of a 6 m × 3.5 m space. This configuration ensures comprehensive monitoring of the entire area, including potential leak points. The collection environment was designed to intentionally generate acoustic signals of pipeline leaks using air compressor equipment on a pipe specimen with 10 leak points, as collecting leak signals by discharging water in actual piping systems poses inherent hazards. Each of the four microphone sensors collected data consisting of 100,000 sample points at a sampling frequency of 100 kHz for a duration of 1 s, with data collection intervals of 5 s. This methodology allows for the capturing of acoustic signals propagated through the air.

For background noise, various mechanical operating sounds around the piping, such as network server equipment and air conditioner indoor unit fan operations, pump motors, and air compressor operations, were measured and collected. The size or range of the leak signal for generating leak signals utilized leak conditions such as leak diameter (H) and air pressure (P). At this time, considering the standard for low-level leaks [28,35], leak signals were measured by varying the pressure between 1 and 2 bars at the location of the leak with a 0.5 mm leak diameter. The data collected in this way were transformed into volume features for each area after applying the data preprocessing techniques introduced in Section 2. Then, they were used for training and performance evaluation of the leak detection model proposed in this study.

### 4.2. Pipe Leak Detection Performance of the Proposed Method

From the data measured and collected in the testbed environment of Section 4.1, a total of 66,000 volume feature data samples were composed, consisting of 33,000 leak volume features and 33,000 background noise volume features, through the data preprocessing process of Section 2. For the training and performance evaluation of the proposed model in Section 3, the data were split into training and test data using a holdout method with a 6:4 ratio. Table 1 presents the training hyperparameters for the model proposed in Section 3. To train each of the two models used in Stage 1, the number of epochs was set to 30, and the batch size was set to 64. Stochastic gradient descent (SGD), which applies momentum, was selected as the learning algorithm. The learning rate was set to $1\times 10^{-3}$, and the loss function was set to categorical cross-entropy loss. On the other hand, epochs, learning algorithm, and loss function for training the proposed model in Stage 2 were the same as in Stage 1, but the batch size was set to 128. In addition, when setting the learning rate, the step decay method was applied to gradually decrease the learning rate from the initial rate of $1\times 10^{-2}$ to $1\times 10^{-3}$ and eventually to $1\times 10^{-5}$ as the number of epochs increased, thereby stabilizing the learning process.

The detailed structure of each domain-specific ResNet-based model presented in Section 3.1, Stage 1, is shown in Table 2. This table provides a step-by-step explanation of the input data processing and the structure of the leak detection model. The model commences with an input layer with dimensions of 40 × 40 × 4, representing the volume feature after the data preprocessing. The first residual block generates an output sized 20×20×16, resulting from two 3×3 convolutional layers, each comprising 16 filters. These layers are responsible for initial data processing and basic feature extraction. The second residual block produces an output with dimensions of 10×10×32, consisting of two 3×3 convolutional layers, each with 32 filters. This block extracts more complex features from the output of the first block, enhancing data representation. The final stage involves a classifier comprising a GAP layer and two fully connected layers. The GAP layer computes the average of each feature map to simplify the data, while the fully connected layers perform the final classification based on these features. The model culminates in a classification into leak and noise classes, achieved through the application of the softmax activation function.

Table 3 presents the results of the leak detection performance comparison of the proposed model based on the application or nonapplication of TL techniques. To prevent overfitting, each model was trained using a five-fold cross-validation method. The performance metrics, such as Precision, Recall, and F1-score, are defined as follows. In this context, the model without applying TL refers specifically to only implementing Stage 2 from the proposed method in Section 3, bypassing Stage 1.
(13)$$Precision=\frac{True\;Positive\;(TP)}{True\;Positive\;\left(TP\right)+False\;Positive\;(FP)}$$
(14)$$Recall=\frac{True\;Positive\;(TP)}{True\;Positive\;\left(TP\right)+False\;Negative\;(FN)}$$
(15)$$F1-Score=2\times \frac{Precision\times Recall}{Precision+Recall}$$

Precision in (13) refers to the proportion of samples that are actual leaks among those classified as leaks by the classification model, Recall in (14) refers to the proportion of actual leak samples that are classified as leaks by the model, and F1-score in (15) represents the harmonic mean of Precision and Recall. Furthermore, Figure 10 shows the confusion matrix for the two models in Table 3, visually representing in a 2D matrix the number of correctly and incorrectly classified cases for leak and normal conditions. As indicated by the results in Figure 10 and Table 3, the model applying TL-based stagewise learning effectively utilized the knowledge of important features of the pretrained input data, resulting in improved leak detection accuracy compared to the model without TL.

Table 4 presents the results of a comparison of the leak detection accuracy performance between the proposed model and existing machine learning and deep learning models. In Table 4, an ensemble of multilayer perceptron (MLP) neural networks, denoted as Ensemble-MLP, independently trained on RMS volume and frequency volume features, showed an accuracy of 97.64%. The ensemble-SVM, which involved extracting 1D features in both time and frequency domains and then independently training and ensembling them using the SVM model, demonstrated an accuracy performance of 86.89%. For this, 1D features were extracted in the time and frequency domains without the last 2D domain mapping block of the data preprocessing structure in Figure 1 and were used to train the SVM model. Similarly, ensemble long short-term memory (LSTM), which is the result of independently training and ensembling LSTM models, showed an accuracy performance of 74.45%.

Furthermore, in Section 3, Stage 1, among the ResNet-based models, the ResNet-RMS utilizing RMS volume features demonstrated an accuracy of 96.10%, and the ResNet-Freq utilizing frequency volume features showed an accuracy of 97.75%. The Ensemble RMS+Freq, which results from ensembling models that independently learned the volume features from both time and frequency domains, exhibited an accuracy of 98.71%. Conversely, the performance of the proposed TL-based model applied in Stage 2 was observed to be the highest, achieving an accuracy of 99.89%.

Figure 11 shows the training performance over epochs for ResNet single models using RMS, frequency pattern volume features, and the proposed model. In Figure 11a, the accuracy of each model is presented for the test dataset over epochs, while Figure 11b shows the loss for the test dataset over epochs. As seen in Figure 11, the proposed model was observed to have performed ensemble learning based on TL, demonstrating superior generalization performance on the test data compared to the ResNet single models.

Finally, to validate the efficiency of the proposed model for real-time detection, it was deployed on a low-specification embedded device, a Raspberry Pi 4, and inference experiments were conducted. The proposed model was optimized using TensorFlow Lite, a lightweight version of the TensorFlow library designed for running AI models on embedded devices. The inference performance was assessed by measuring the time taken to classify leaks and normal conditions following data preprocessing, employing the same method as the originally proposed approach. The dataset utilized in the inference process comprised 25 leaks and 25 normal instances, totaling 50 instances. The inference results, as shown in Figure 12, confirmed that the average inference time for these 50 consecutive instances was 0.6974 s, well within 1 s, demonstrating that the proposed model is suitable for real deployment environments.

### 4.3. TL Performance of the Proposed Method

Section 4.2 addresses the training task involving model transfers between different environments in the source domain and the target domain, in contrast to Section 4.1. To collect source domain data, a testbed environment was established within a space measuring 17 m in width and 14 m in length, as shown in Figure 13. The same commercial microphone acoustic sensors used in Section 4.1 were utilized to collect leak signals. Four commercial microphone acoustic sensors were strategically placed at each corner of the testbed to consider shadow areas and eliminate blind spots.

The leak data collection environment in the source domain considered 25 distinct leak conditions at 100 locations to gather the diverse data types necessary for transfer learning. Specifically, the leak conditions applied to each location included five leak diameters, with a range of 0.5–2.5 mm at 0.5 mm intervals. In addition, pressure conditions were set in five cases, ranging from 1 bar to 5 bars at 1 bar intervals. Consequently, leak data was collected under 25 leak conditions, combining the five leak diameters and five pressure conditions at each location.

The data collected from the aforementioned experimental environment were transformed into a total of 66,000 RMS volume features and 66,000 frequency volume features through the same data preprocessing process described in Section 2 and were used for model training in the source domain. As a result of the training, the ResNet model using RMS volume features in the time domain demonstrated a leak detection accuracy of 99.79%, while the ResNet model using frequency volume features showed an accuracy performance of 99.42%. These pretrained models were then transferred as base models for the ensemble model training in Stage 2 of Figure 4. Therefore, the experimental objective in this section was to examine the leak detection performance of the ensemble model trained in the target domain using the data collected in Section 4.1, with models pretrained in the source domain.

Figure 14 demonstrates a comparison of the impact of applying TL on model accuracy with varying amounts of training datasets. When the entire dataset is used, it is observed that the accuracy performance of both models is nearly similar. In addition, up to about 25% (10,000 samples) of the total dataset, both models maintained stable performance without significant decreases in accuracy. However, when the dataset size was reduced to about 13% (5000 samples), a gradual significant decline in performance could be observed in the model without the TL application. In contrast, the proposed ensemble model with TL applied maintained an accuracy above a certain level even with a very small dataset size of approximately 2.5% (1000 samples). These results demonstrate the advantage of the proposed TL-based ensemble model in enabling effective learning with a limited amount of leak data and maintaining high performance even in situations where collectible data in the target domain are restricted.

## 5. Conclusions

This paper proposes an automatic weight redistribution ensemble model using TL-based feature fusion from multiple sensors for microleak detection. The time series acoustic data of leaks and noise collected from multiple microphone sensors were transformed into 2D RMS and frequency pattern features suitable for CNN-based model training. These transformed multiple 2D pattern features from each area were then combined in depth to form RMS and frequency volume features. RMS and frequency volume features were independently trained on a ResNet-based model for TL in Stage 1. Subsequently, in Stage 2, the models pretrained in Stage 1 were transferred and ensembled, and the ensemble weights of the model were automatically redistributed during the training process.

The experimental results confirmed that the performance of the proposed model was superior to the traditional machine learning and deep learning methods considered in this study. Notably, even when trained with a limited dataset in cases where the source and target domains were different, the proposed model maintained high accuracy, demonstrating the advantages of the TL-based ensemble learning approach in this research. Ultimately, it was verified that the proposed model could classify leak and noise signals with reliable accuracy performance by adaptively assigning ensemble weights using pretrained knowledge of each domain, even when the amount of data available for model training was very limited.

## Figures and Tables

**Figure 1 sensors-24-04999-f001:**
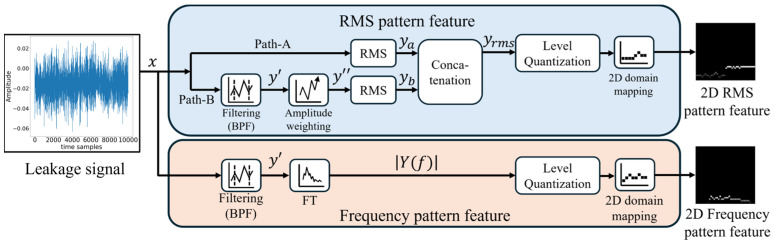
Data preprocessing diagram for training the proposed model [28].

**Figure 2 sensors-24-04999-f002:**
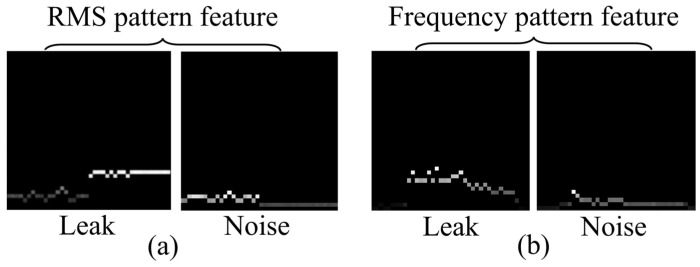
Root-mean-square (RMS) and frequency pattern features for leak signal and noise: (**a**) RMS pattern feature, (**b**) Frequency pattern feature.

**Figure 3 sensors-24-04999-f003:**
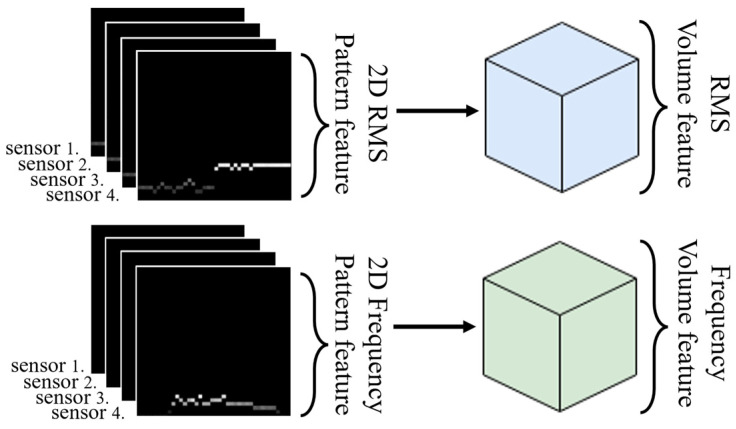
Root-mean-square (RMS) and frequency volume features.

**Figure 4 sensors-24-04999-f004:**
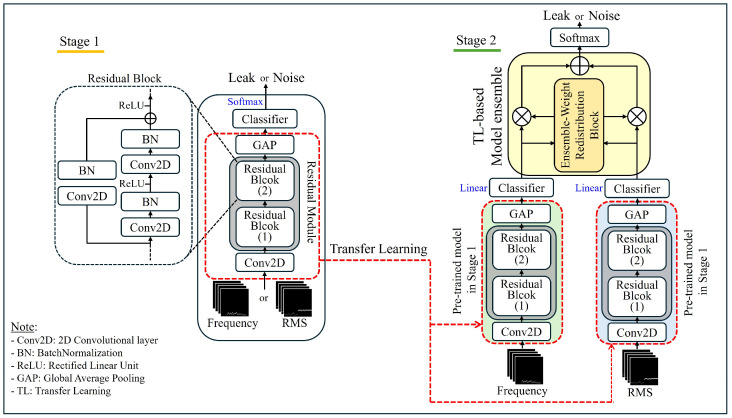
Proposed method architecture.

**Figure 5 sensors-24-04999-f005:**
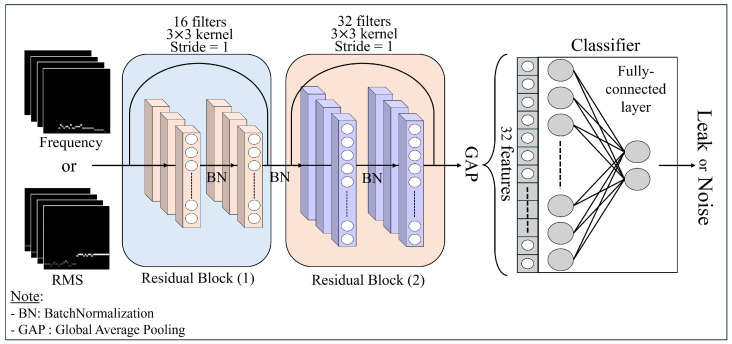
Stage 1 of single model architecture for each domain.

**Figure 6 sensors-24-04999-f006:**
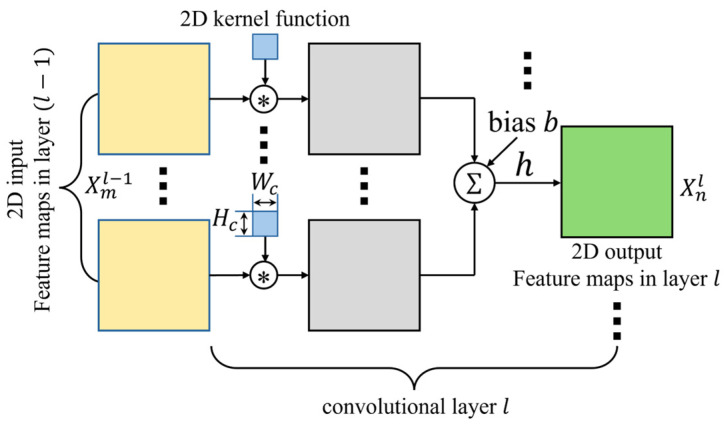
Convolutional operation process using kernel function for the 2D CNN.

**Figure 7 sensors-24-04999-f007:**
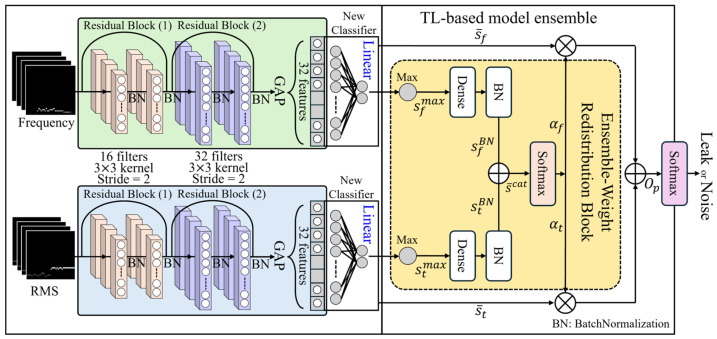
Stage 2 of the proposed method architecture.

**Figure 8 sensors-24-04999-f008:**
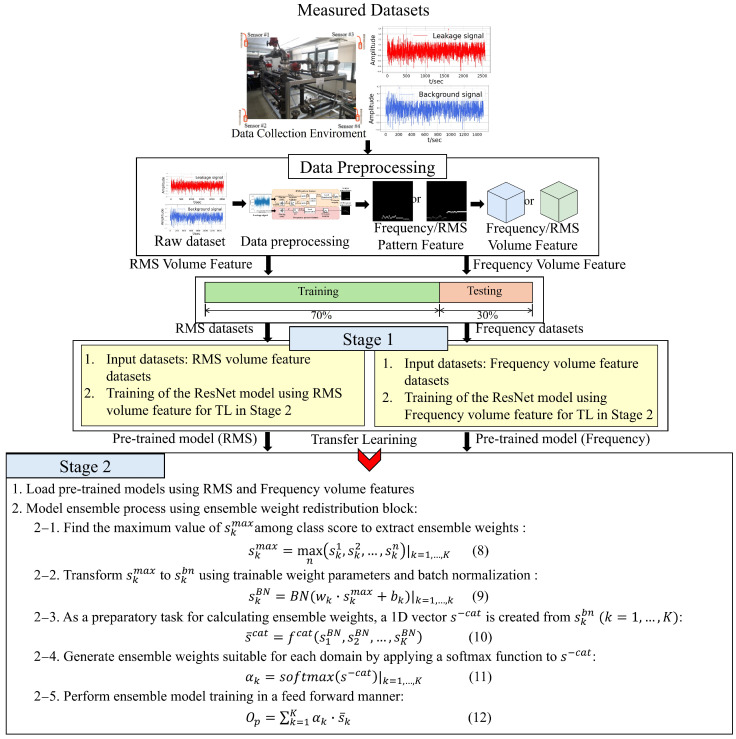
Diagram of the proposed leak detection algorithm based on ensemble learning with a phased model transfer approach.

**Figure 9 sensors-24-04999-f009:**
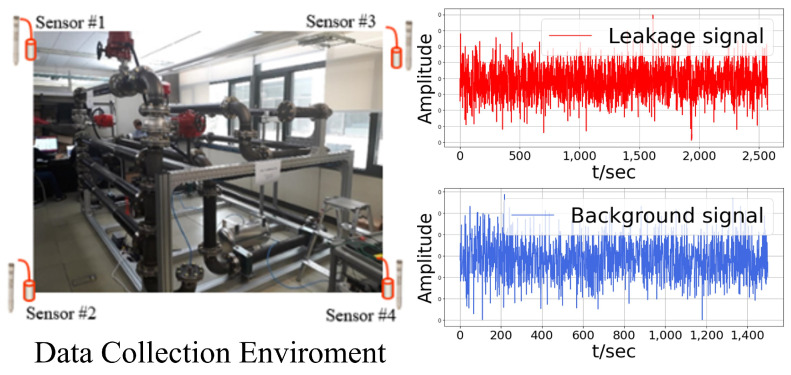
Plant piping testbed environment for leak signal measurement and collected acoustic signals (leak, noise).

**Figure 10 sensors-24-04999-f010:**
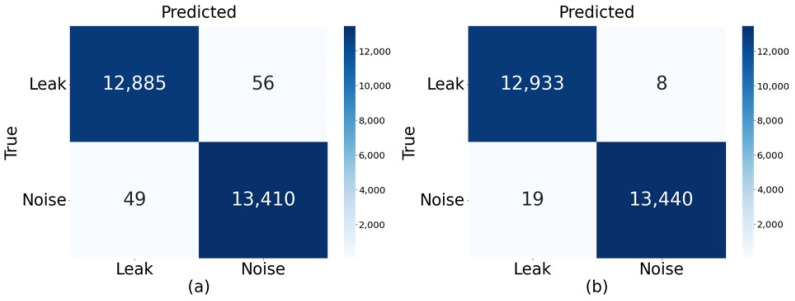
Confusion matrix results of the proposed model according to the application of TL techniques: (**a**) without the application of TL, (**b**) with the application of TL.

**Figure 11 sensors-24-04999-f011:**
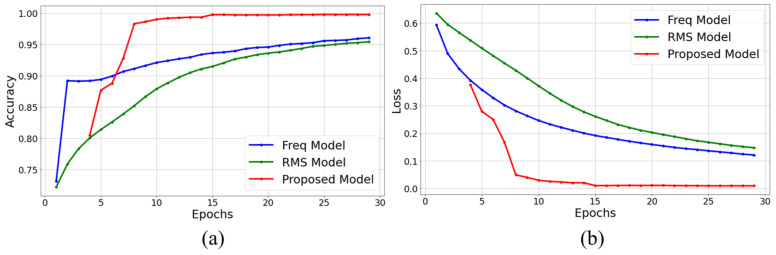
Analysis of test accuracy and test loss for the proposed method and models without an ensemble: (**a**) epochs versus test accuracy, (**b**) epochs versus test loss.

**Figure 12 sensors-24-04999-f012:**
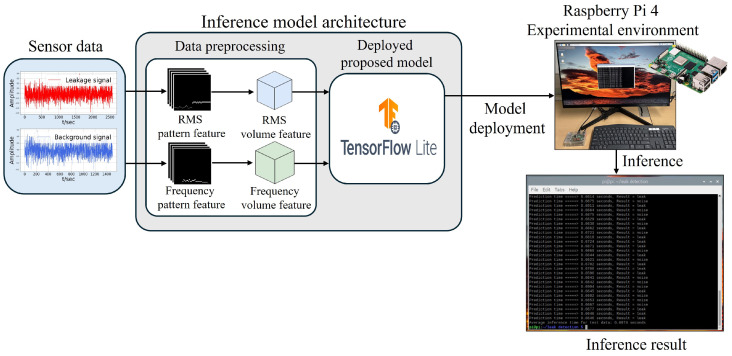
Inference process of the proposed model on Raspberry Pi 4.

**Figure 13 sensors-24-04999-f013:**
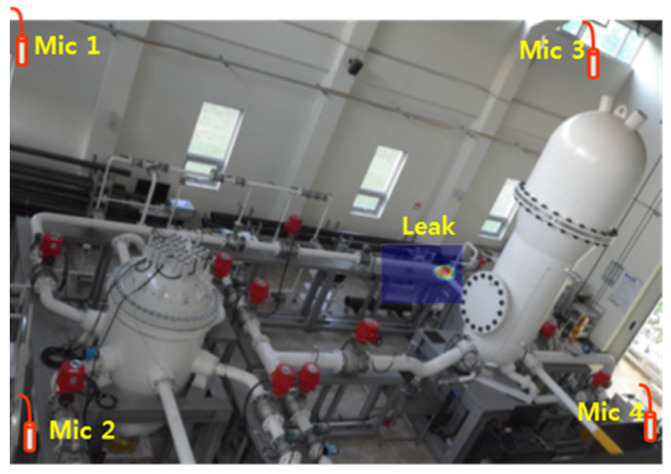
Testbed environment for source domain data collection [8].

**Figure 14 sensors-24-04999-f014:**
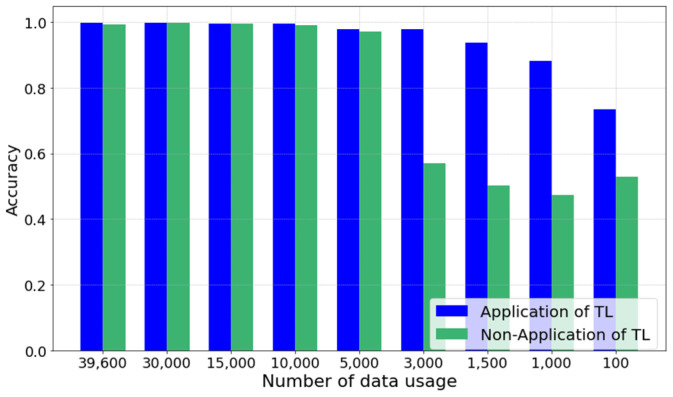
Comparison of leak detection accuracy performance relative to the number of training data samples using the target domain.

**Table 1 sensors-24-04999-t001:** Hyperparameter settings for the proposed model training in Stages 1 and 2.

Parameters	ResNet (RMS)	ResNet (Freq)	Ensemble Model
Epochs	30	30	30
Batch size	64	64	128
Optimizer	SGD + Momentum		
Learning rate	$1\times 10^{-3}$	$1\times 10^{-3}$	($1\times 10^{-2}$, $1\times 10^{-3}$, $1\times 10^{-5}$)
Loss	Categorical cross-entropy		

**Table 2 sensors-24-04999-t002:** Detailed structure of a single ResNet model.

Stage	Output Size	Layer
Input	40×40	Input Layer
Residual Block (1)	20×20	$\left[\begin{array}{cc} 3\times 3, & 16\\ 3\times 3, & 16 \end{array}\right]$
Residual Block (2)	10×10	$\left[\begin{array}{cc} 3\times 3, & 32\\ 3\times 3, & 32 \end{array}\right]$
Classifier	2	Global average pool, FC: [32, 2] Softmax
# params.	18,930	

**Table 3 sensors-24-04999-t003:** Evaluation of pipe leak detection accuracy (percentage) using various performance metrics.

Methods	Proposed Method without TL	Proposed Method with TL
Accuracy	99.60	99.89
Precision	99.68	99.94
Recall	99.64	99.86
F1-score	99.66	99.90
Reference	Only Stage 2	Stage 1 + Stage 2

**Table 4 sensors-24-04999-t004:** Performance comparison of the low-level pipe leak detection accuracy.

Methods	Ensemble-MLP	Ensemble-SVM	Ensemble-LSTM	-
Accuracy [%]	97.64	86.89	74.45	-
Methods	ResNet-RMS	ResNet-Freq	Ensemble-RMS+Freq	Proposed Method
Accuracy [%]	96.10	97.75	98.71	99.89

## Data Availability

The data presented in this study are available upon request from the corresponding author. The data are not publicly available due to privacy concerns.

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
