# Peer review of "Automatic Weight Redistribution Ensemble Model Based on Transfer Learning to Use in Leak Detection for the Power Industry"

_sensors, 2024, doi:10.3390/s24154999_

Round 1

Reviewer 1 Report

Comments and Suggestions for Authors

The author used an automatic weight redistribution ensemble model based on transfer learning to detect leaks in diverse power plant environments, and specific suggestions are as follows:

1. The writing and arrangement of papers should be more rigorous. For example, the parameter symbols contained in formula 1 are not consistent with the writing in the text. Tables 1, 2, and 3 have exactly the same header names. The first paragraph on page 12 describes "Figure 12 shows the confusion matrix for the two models in Table 3", but the confusion matrix figure is Figure 11. As for the numbering of the figure, Figure 2 is followed by Figure 4, obviously wrong. The author should check and revise the article carefully.

2. In the paper, the authors described that the proposed model can detect pipeline low leakage through multimode feature fusion. If only the microphone sensors are used to collect signals, it is called multimode, is it appropriate?

3. The results in Figure 2 do not support the conclusions and effects described in the text.

4. In the first residual block, convolutional layers are configured to output 16 feature maps by applying a 3×3 kernel filter. In the second residual block, the convolution layers are configured to output 32 feature maps. Why does the article choose these parameters and not others?

5. In Figure 5, there is a special introduction to Residual Block (2), why is it not described in the text?

6. The writing logic in Section 3.1 is a bit messy and does not clearly reflect the execution of Stage1.

7. When applied to fault feature detection, the performance of the microphone sensor will be limited by the installation distance, and the paper does not clearly explain the installation position of the sensor in the test platform. In addition, applied to the power plant environment, the installation of sensors has certain standards and requirements. Is the fault diagnosis method introduced in this paper limited by application scenarios?

8. The parameters in Table 2 are introduced in detail, but the contents in Table 1 are not described.

9. As shown in Table 3, there is little difference in leak detection performance between the two methods. The method without TL only uses Stage 2, so is the training time faster than the method with TL? If the processing time of former is shorter, then the “reducing training time and enhancing performance” mentioned on the second page of the paper is inconsistent with this conclusion.

In summary, although the method proposed in this paper has been verified by experiments, there are many mistakes in this paper, and the research significance is not clear, so it is recommended to reject the paper.

Comments on the Quality of English Language

English language is okay.

Author Response

We would like to express our gratitude for taking the time to review our manuscript, titled 'Automatic Weight Redistribution Ensemble Model Based on Transfer Learning for Leak Detection in the Power Industry' (Manuscript ID: sensors-3096704). Your valuable feedback has enabled us to make significant improvements to our work.

Thanks to your detailed and insightful comments, we were able to identify and correct several major errors throughout the manuscript. Your input has been instrumental in enhancing the clarity, accuracy, and overall quality of our paper. We appreciate your thorough review and believe that the revised manuscript is much stronger because of your constructive criticism. The responses to the review have been written in a Word file and submitted.

Thank you once again for your time and effort.

Reviewer 2 Report

Comments and Suggestions for Authors

This study proposes an automatic weight redistribution ensemble model based on transfer learning to detect leaks in various power plant environments, addressing the limitations of site-specific AI methods. The model processes time series acoustic data from multiple sensors into three-dimensional multimode root-mean-square and frequency volume features for accurate leak detection. The results show that this method effectively distinguishes low-level leaks and noise compared to existing techniques, even with limited training data. This study is straightforward; however, the following comments should be addressed before this manuscript can be considered for publication.

1.      Although the proposed model adopts transfer learning and ensemble techniques (line 69), the author needs to illustrate the novelty of this study since these methods have already been proposed and used in several studies.

2.      The authors mentioned that the model can provide accuracy and efficiency for leak detection (line 108). However, the detailed analysis of the computational requirements of the proposed model in Figure 5 is not provided in this manuscript. Due to the complexity of training residual networks and an ensemble approach, the authors should provide basic model complexity analysis.

3.      For the data acquisition procedure, the authors should provide more details about the experiments. For instance, the dimensions of the testbed environment, sensor types, and details of the collection process should be provided.

4.      The experimental results demonstrate exceptionally high detection accuracy. However, the paper does not adequately address how the authors mitigate the risk of overfitting in the proposed model and during the validation process.

5.      For transfer learning, the paper should provide more specifics on how the models in stage 1 can be adapted to stage 2. Specifically, what criteria and rationale are used to create the model architecture in stage 2 based on stage 1?

Author Response

We would like to thank you for taking the time to review our manuscript, allowing us to resubmit our manuscript entitled “Automatic Weight Redistribution Ensemble Model Based on Transfer Learning to Use in Leak Detection for the Power Industry” (Manuscript ID: sensors-3096704). Moreover, we would like to thank you for your feedback, which enabled us to create revisions to improve our manuscript.  The responses to the review have been written in a Word file and submitted.

Round 2

Reviewer 1 Report

Comments and Suggestions for Authors

The authors have implemented significant revisions to the manuscript and provided a comprehensive response to the comments.

Reviewer 2 Report

Comments and Suggestions for Authors

By carefully checking the revised manuscript and corresponding response, the comments proposed by the reviewer are addressed appropriately. 

From the reviewer's perspective, the content quality is significantly improved and the manuscript is recommended for publication in its present form.